# Multiple membrane extrusion sites drive megakaryocyte migration into bone marrow blood vessels

Edward Brown[1], Leo M Carlin[2,3], Claus Nerlov[4], Cristina Lo Celso[5,6], Alastair W Poole[1]

Platelets, cells central to hemostasis and thrombosis, are formed from parent cell megakaryocytes. Although the process is highly efficient in vivo, our ability to generate them in vitro is still remarkably inefficient. We proposed that greater understanding of the process in vivo is needed and used an imaging approach, intravital correlative light electron microscopy, to visualize platelet generation in bone marrow in the living mouse. In contrast to current understanding, we found that most megakaryocytes enter the sinusoidal space as large protrusions rather than extruding fine proplatelet extensions. The mechanism for large protrusion migration also differed from that of proplatelet extension. In vitro, proplatelets extend by sliding of dense bundles of microtubules, whereas in vivo our data showed the absence of microtubule bundles in the large protrusion, but the presence of multiple fusion points between the internal membrane and the plasma membrane, at the leading edge of the protruding cell. Mass membrane fusion, therefore, drives megakaryocyte large protrusions into the sinusoid, significantly revising our understanding of the fundamental biology of platelet formation in vivo.

## Introduction

Platelets are small anucleate blood cells with principal roles in hemostasis and thrombosis. They are formed from large precursor cells, megakaryocytes, by a highly efficient process in vivo, generating $10^{11}$ platelets per day in adult humans (1). Mature megakaryocytes have a large amount of internal membrane (2, 3, 4), allowing a single megakaryocyte to generate approximately 4,000 platelets. From its perisinusoidal niche within the bone marrow, the megakaryocyte extends projections (5), which are thought to fragment to form platelets (6, 7, 8), into the vasculature.

It is generally thought that these projections are predominantly fine proplatelet extensions. The reason for this is that in vitro megakaryocytes, in contact with fibrinogen- or fibronectin-coated surfaces, do indeed break up into multiple thin, tubular, and bifurcating proplatelet extensions (9, 10, 11). Elongation of the proplatelet in vitro is driven by a dynein-dependent sliding of overlapping cortical microtubule (MT) bundles that line the length of the proplatelet shaft (12). However, the final stage process required for platelet generation is release of platelets from proplatelet extensions, and this has very rarely been observed experimentally. Attempts to generate platelets in vitro have yet to yield large numbers, and a huge disparity exists between the efficiency of platelet production in vivo and in vitro (13, 14). We propose that platelet generation in vivo needs to be better understood to allow us to rationally refine current in vitro models so that their efficiency in platelet generation may be enhanced.

In contrast to the proplatelet model, a recent publication observed that at least in mice "stressed" to up-regulate platelet production, by inducing acute thrombocytopenia, substantial proportions of the megakaryocyte or even whole cells may exit the marrow into the sinusoidal space as large protrusions (15). This is consistent with another recent study by Lefrançais et al (16) who showed that approximately 50% of platelet production occurs not in the bone marrow but in the lung, suggesting that whole megakaryocytes or large fragments of these cells may migrate from bone marrow to the lung.

Intravital imaging has previously shown megakaryocyte-derived structures entering bone marrow sinusoids (5); however, the limited resolution of intravital light microscopy cannot reveal fine structural details or cytoskeletal organization. EM has previously been used to visualize in much greater detail megakaryocyte projections into sinusoidal space (17, 18, 19), and large protrusions have been shown by this approach to be distinct from proplatelets and primarily found during thrombocytopenia when platelet demand is increased (15).

The key points we wished to address in this study, therefore, were whether megakaryocytes in "non-stressed" (non-thrombocytopenic) and stressed mice projected into marrow sinusoidal vessels as fine proplatelet extensions or large protrusions and, importantly, what the cellular mechanism underlying these projections is. The data reveal

---

[1]School of Physiology and Pharmacology, Faculty of Medical and Veterinary Sciences, University of Bristol, Bristol, UK   [2]Cancer Research UK Beatson Institute, Garscube Campus, Glasgow, UK   [3]Inflammation, Repair, and Development, National Heart and Lung Institute, London, UK   [4]MRC Molecular Hematology Unit, Weatherall Institute of Molecular Medicine, University of Oxford, John Radcliffe Hospital, Oxford, UK   [5]Department of Life Sciences, Faculty of Natural Sciences, Imperial College London, London, UK   [6]The Francis Crick Institute, London, UK

Correspondence: a.poole@bristol.ac.uk

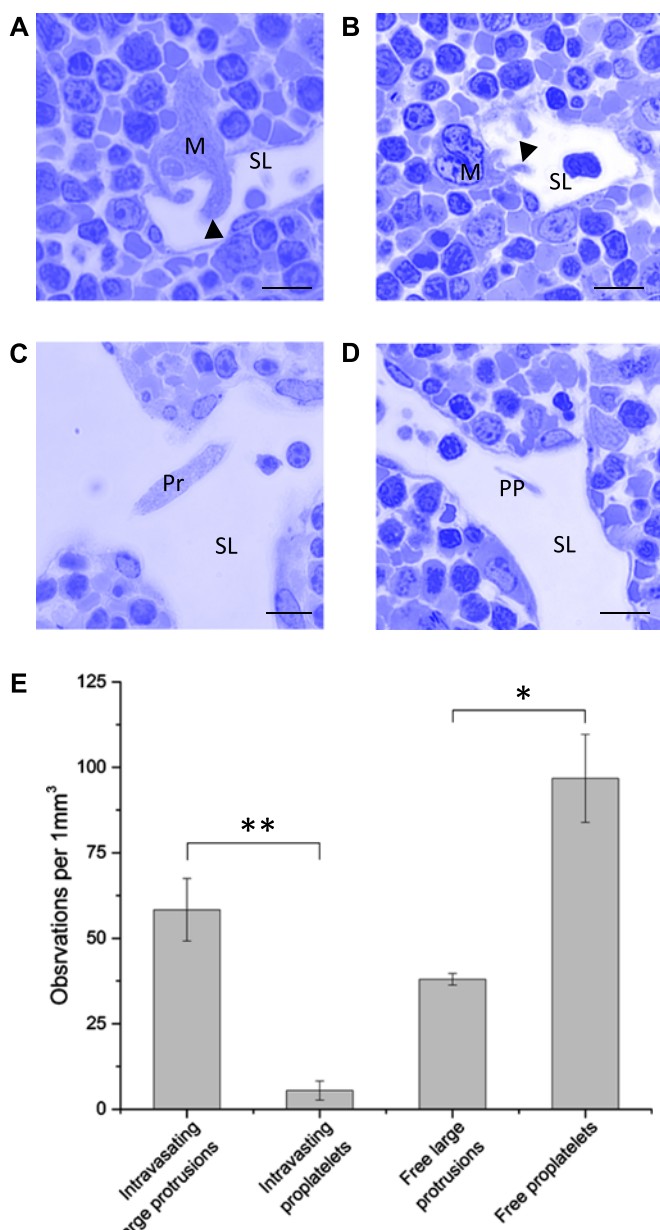

**Figure 1. Megakaryocytes predominantly enter the circulation as a large protrusion.**
**(A)** Megakaryocyte (M)-intravasating large protrusion (arrowhead) extending into sinusoid lumen (SL). **(B)** Megakaryocyte (M)-intravasating proplatelet (arrowhead) extending into SL. **(C)** Free large protrusion (Pr) within SL. **(D)** Free proplatelet within SL. Scale bars represent 10 μm. **(E)** Observations of intravasating large protrusions, intravasating proplatelets, free large protrusions, and free proplatelets per 1 mm³ of bone marrow. Bone marrow of the diaphysis region obtained from flushed femurs. Data presented are mean ± SEM. **$P$ < 0.005. *$P$ < 0.01 (unpaired two-tailed $t$ test, three independent experiments).

the substantial predominance of large protrusions as the exit route for megakaryocyte projections in non-thrombocytopenic mice and also reveal a new mechanism of cellular protrusion and migration for megakaryocytes, involving multiple fusions between internal and external membrane systems at the leading edge of the cell.

# Results

## Megakaryocytes predominantly enter sinusoids as large protrusions

It was important first to quantify the classes of megakaryocyte projections into marrow sinusoids. Serial histological sections from whole murine femurs were, therefore, assessed for proplatelet and large protrusion content (Figs 1A–D and S1). Surprisingly, this revealed very few classical proplatelet extensions resembling those seen in in vitro–cultured megakaryocytes. To quantify the abundance of these structures, we defined proplatelets as having a diameter ≤3 μm, whereas structures with a diameter >3 μm were classed as large protrusions. Quantification revealed that >90% of megakaryocytes were entering the circulation as large protrusions (58.3 observations per 1 mm³ of bone marrow [Fig 1E]) compared with proplatelets (5.5 observations). Once detached from the megakaryocyte cell body, however, there was a shift toward a proplatelet morphology, with 72% of free structures being proplatelets.

## Large protrusions and proplatelets differ in morphology and MT arrangement

Proplatelets formed from megakaryocytes in vitro (Fig 2A and B, and Video 1) and those found in histological sections (Fig 1B and D) have a typical diameter of between 2 and 3 μm and often possess bulges along their length and at their ends, thought to be nascent platelets that will subsequently be budded from the extension. Electron tomography of the tip of an extending proplatelet, formed in vitro from a megakaryocyte, revealed a dense cortical band of MTs (Fig 2C) just under the plasma membrane, similar to that of a platelet (20). 42% of total MT content is found within 50 nm of the plasma membrane of the leading edge of proplatelets and predominantly in tight bundles (Fig 2D). Large protrusions are not typically formed from megakaryocytes in vitro, suggesting that some element of the bone marrow microenvironment is crucial for their generation. From histological sections (Fig 1A and C), the diameter of large protrusions typically ranged from 4 to 10 μm (Fig 2E and Video 2), without the bulges seen in proplatelets. Tomography of the large protrusion tip (Fig 2F), formed in vivo, revealed a more homogenous distribution of MTs with fewer dense bundles. There were also fewer MTs (14% total) within 50 nm of the plasma membrane (Fig 2G) and no accumulation at the leading edge. The data, therefore, demonstrate that large protrusions, formed in vivo, are structurally distinct from proplatelets, with a different MT arrangement and likely to have a different mechanism of extension.

## Intravital correlative light electron microscopy (CLEM) of bone marrow

Although electron micrographs provide high enough resolution to reveal the structural details of megakaryocytes undergoing thrombopoiesis, they only represent snapshots of a dynamic process. For this reason, we have developed a form of intravital CLEM, which allows real-time live observation of thrombopoiesis in vivo combined with high-resolution transmission electron

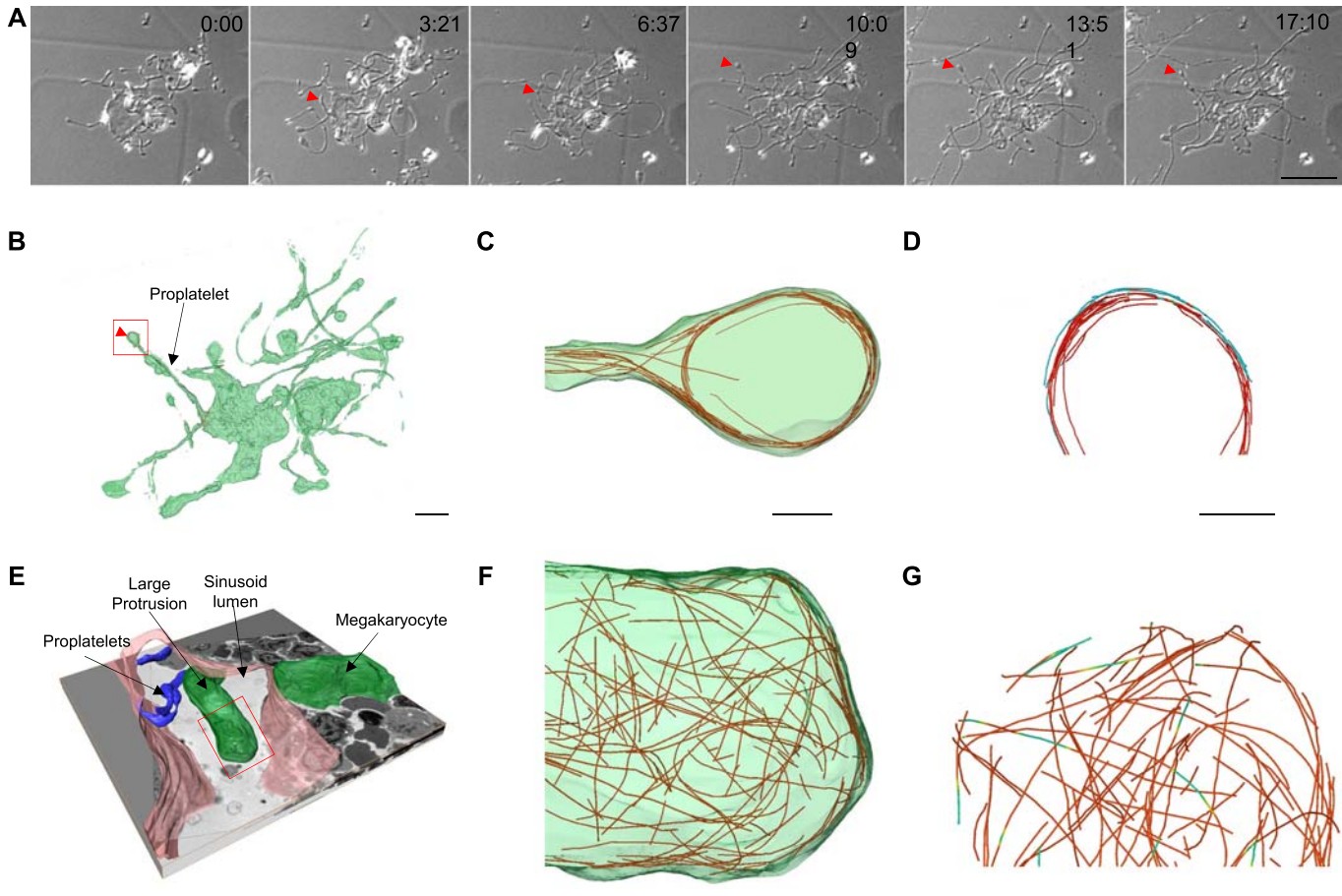

**Figure 2. Proplatelets and large protrusions are morphologically distinct and possess different MT arrangements.**
**(A)** Selected images from time-lapse DIC microscopy of proplatelet formation in vitro. One proplatelet tip has been followed (arrowhead). **(B)** Model constructed from TEM section of the same megakaryocyte (green). **(C)** MT (red) arrangement within the proplatelet tip. **(D)** MTs at the leading edge of the proplatelet. MTs within 50 nm of the plasma membrane are colored blue. **(E)** Model of large protrusion constructed from serial TEM sections showing the megakaryocyte (green) protrusion entering the sinusoid (pink). Free proplatelet-like structures (blue) are also present. **(F)** MT (red) arrangement within the protrusion tip. **(G)** MTs at the leading edge of the protrusion. MTs within 50 nm of the plasma membrane are colored blue. Scale bars represent 50 μm in (A), 10 μm in (B, E), and 1 μm in (C, D, F, G). Bone marrow of the diaphysis region obtained from flushed femurs. Images representative of three independent experiments.
Source data are available for this figure.

microscopy (TEM) and large-volume electron tomography to reveal structural detail of an event that we observe in the mouse. Details of the intravital CLEM process are illustrated in Fig S2 and the associated Video 3.

### Intravital CLEM microscopy of megakaryocyte extrusion into blood vessel sinusoids

Representative intravital fluorescence microscopy of vWF-tdTomato–expressing bone marrow showed two extensions into a blood vessel sinusoid from a megakaryocyte resident in the marrow space (Fig 3A and Video 4). At the resolution afforded by light microscopy in vivo, it was not possible to determine whether these extensions were likely to be fine proplatelet ones or large protrusions. Both extensions are dynamic structures, and protrude and retract within the sinusoid lumen over the 20-min movie but did not appear to do so in a coordinated manner, suggesting that blood flow alone was not responsible for their

extension. It was important to visualize the extensions then at TEM resolution level, and we were encouraged to find that the model constructed from serial TEM sections (Fig 3B and Video 5) shows protrusions that are similar to those seen in the last frame of live imaging, suggesting there has been little structural change in the period between live imaging and fixation of the bone marrow. The orientation of the longer protrusion has, however, altered, likely because of cessation of blood flow within the sinusoid lumen on euthanizing the mouse. The resolution afforded by EM allowed us to see that the two protrusions observed by intravital microscopy originate from two separate megakaryocytes (Fig 3B, M1 and M2). EM-level resolution also revealed an additional small protrusion that is approximately 4 μm in length, arising from a third megakaryocyte penetrating the endothelium (Fig 3C).

Large-volume electron tomography performed at the neck region (i.e., the region where the protrusion connects to the cell body) of the longest protrusion (extending from M2) revealed a large pool of internal membrane system (IMS) and MTs in close proximity to

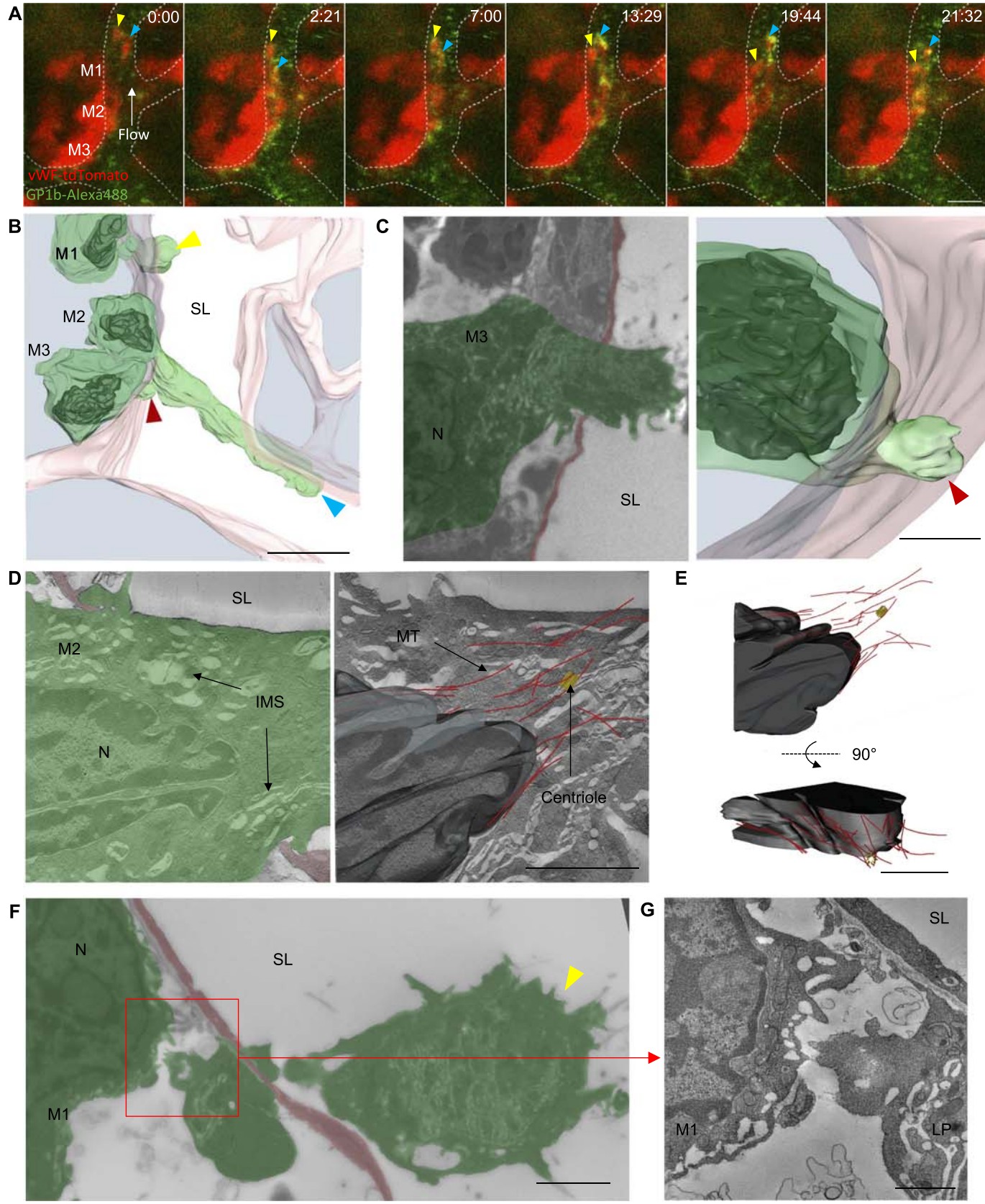

a centriole (Fig 3D). The MTs were organized in a conical arrangement around the tip of the protruding nucleus (Fig 3E). Detailed inspection of the smaller protrusion associated with M1 showed it to be completely separate from its parent megakaryocyte (Fig 3F and G). This is the first time release of megakaryocyte material into circulation has been followed in vivo and shown with EM resolution, and interestingly the point of scission was not on the luminal side but within the parenchyma of the bone marrow. This provides additional evidence that shear forces alone, derived from flowing blood, are likely not to be solely responsible for scission because it would otherwise be expected that the point of scission would lie in the lumen of the vessel. This evidence, together with the observation noted above that large protrusions extend and contract in the sinusoid, suggests a different, and probably regulated, mechanism for separation of these large protrusions. The observation that the arrangement of MTs in the protrusion is not bundled in a sub-plasma membrane location also suggests a mechanism of protrusion distinct from that seen for proplatelet extension in vitro.

## Megakaryocytes extend multiple protrusions that anchor to the endothelium

Analysis of serial histological sections of bone marrow revealed that megakaryocytes frequently extend multiple protrusions into the sinusoid lumen simultaneously (Fig 4A). In fact, 52% of the intravasating megakaryocytes we observed possessed more than one protrusion, with as many as four separate protrusions from one megakaryocyte being recorded. Electron tomography of exit points of protrusions shows them to anchor to the luminal side of the endothelium. This is achieved by folding over of excess membrane (Fig 4B and C) or insertion of filiopodia-like structures into the luminal surface of endothelial cells (Fig 4D and E), forming a tight interdigitation between the two cells and consequently an anchor point at the exit site from the endothelium. This anchoring is likely to be important to allow force to be applied to extrude the protrusion into the sinusoidal space.

## Loss of peripheral zone allows fusion between IMS and plasma membranes

It was important to determine the mechanism underlying extrusion of large protrusions into the sinusoid because the evidence provided here suggests this is likely not parallel to in vitro proplatelet extension. In non-protruding megakaryocytes, electron tomography demonstrated a clearly defined and characteristic peripheral zone present at the rear and front of the cell (Fig 5A). This region was approximately 0.5–2 μm wide and largely devoid of organelles or IMS and has been reported in previous publications (4, 21). However,

in protruding megakaryocytes, this peripheral zone was absent at both the rear and front of the cell (Fig 5B). With no zone to form a boundary around the IMS, it is then free to move into apposition with the plasma membrane. No evidence of extensive actin cytoskeleton was observed at either the front or rear of protruding megakaryocytes.

Furthermore, large-volume tomography of the tip of the protrusion (Fig 6A and Videos 2 and 6) showed the IMS to be far less densely packed than in non-protruding megakaryocytes, often presenting in the form of large vacuoles connected by narrow tubules (Fig 6B). This reduction in packing density of the IMS would be expected if the membrane were being extruded into the plasma membrane. Although other membrane compartments such as vesicles, rough ER, and Golgi were present at the protrusion tip (Fig 6C), these were not continuous with the IMS. Crucially, the entire IMS within this reconstructed volume was shown to be a single continuous membrane and, importantly, there were numerous points of fusion with the plasma membrane (Fig 6D and E), effectively making the two membranes continuous. MTs were also commonly present at points of membrane fusion (Fig 6F), suggesting a possible role in trafficking IMS to these sites. In summary, the data implicate a highly distinct mechanism for megakaryocyte migration into the sinusoidal vessel space, where mass fusion occurs between internal and external membranes, driving protrusion extension, guided by MT structures.

## Protrusion extension is associated with increasing plasma membrane surface area

Finally, if megakaryocyte protrusions were migrating through the endothelium by conventional cellular locomotion, i.e., extending the leading edge and retracting the rear, there would be no change in overall surface area. If, however, extension of a protrusion was achieved by mass membrane fusion of internal and plasma membranes at the tip and assuming there was no internalization of membrane elsewhere at an equivalent rate, there would be a resultant increase in total cell surface area. To determine whether this was the case, intravital microscopy was performed on the calvarium of vWF-tdTomato–expressing mice to obtain z-stacks of protruding megakaryocytes at intervals over a 30-min period. As protrusion events are rare, to obtain sufficient data for this part of the study, mice were platelet-depleted 4 days before imaging to increase the rate of platelet production and hence the number of observable protrusions. The models generated from the z-stacks (Fig 7A and Video 7) allowed us to calculate protrusion length and cell surface area over time (Fig 7B). These data show a positive correlation between surface area and protrusion length (Fig 7C). Therefore, IMS extrusion at the leading edge of the protrusion is likely to be the predominant mechanism by which megakaryocytes

**Figure 3. Megakaryocyte large protrusions can be observed by intravital correlative EM.**
**(A)** Selected images from time-lapse intravital microscopy showing three megakaryocytes expressing vWF-tdTomato (red) (M1–M3) adjacent to a sinusoid and two proplatelet-like structures (yellow and blue arrowheads) positive for glycoprotein 1b (green) within the lumen. **(B)** 3D model generated from a stack of TEM images of the area observed in (A). The three megakaryocytes (green with black nuclei) adjacent to the sinusoid (pink) are all extending protrusions into the sinusoid lumen (SL). **(C)** Single TEM image and 3D model of M3. **(D)** Single tomographic slices showing the IMS, MTs (red), and a centriole (yellow). **(E)** 3D model of the nucleus (black), MTs, and centriole. **(F)** Single TEM image of M1 showing the detached large protrusion. **(G)** Single tomographic slice from boxed region in (F) of the interface between M1 and the detached large protrusion. Bone marrow from intravital CLEM of calvarium. Scale bars represent 25 μm in (A, B); 5 μm in (C, F); and 2 μm in (D, E, G). Images representative of a total of six megakaryocytes from three independent experiments.
Source data are available for this figure.

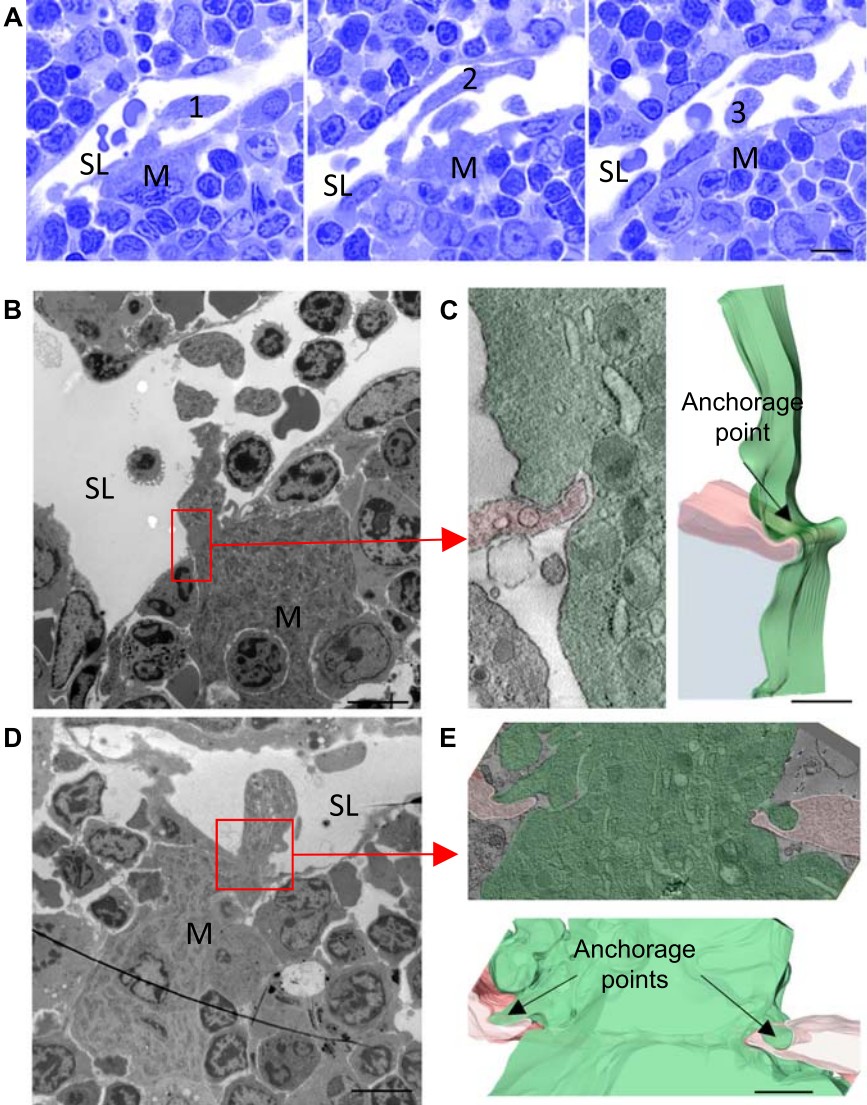

**Figure 4. Megakaryocytes can produce multiple large protrusions simultaneously that anchor into the endothelium.**
**(A)** Histological sections of a single megakaryocyte (M) extending three separate large protrusions into the sinusoid lumen (SL). **(B, D)** TEM images of protruding megakaryocytes, highlighting an area selected for tomography (red box). **(C, E)** Single tomographic slices and respective models of the protrusion membrane (green) overlapping or inserting into the luminal surface of the endothelium (pink). Bone marrow of the diaphysis region obtained from flushed femurs. Scale bars represent 10 µm in (A); 5 µm in (B, D); 500 nm in (C); and 1 µm in (D). Images representative of five independent experiments. Source data are available for this figure.

extend through endothelium into marrow sinusoids. Similar results were observed with three other protrusions from independent experiments (Fig S3A–C).

## Discussion

In this study, we were interested to understand the mechanistic detail of platelet generation in vivo, in part to attempt to understand the discrepancy between the efficiency of platelet production in the body and that of current in vitro settings. Although there have been studies previously conducted to visualize platelet generation in vivo, the resolution and magnification of in vivo light microscopy was limiting in being able to provide details of the process. In particular, we were interested to understand whether the currently accepted model proplatelet formation in vitro pertained also to the in vivo setting, and because of the small size of

these structures, we decided that more details were required than could be provided by in vivo light microscopy.

We, therefore, developed an approach to visualize megakaryocytes in the bone marrow combining light microscopy and EM, which we have termed intravital CLEM. Using this approach, combined with detailed ultrastructural determination by large-volume electron tomography, the data we have generated challenge the current accepted mechanism of proplatelet extension. Although proplatelets are clearly present within the bone marrow sinusoids, they are not the predominant structure extending from the bone marrow parenchyma, and the driver for extension is not a product of MT sliding. Rather, megakaryocytes predominantly extend large protrusions into the blood vessel space by a mechanism that involves mass fusion between the internal and external membranes. The observation that the megakaryocyte anchors itself to the endothelium at the point of exit supports the idea that it is static at this exit site and, therefore, that membrane extrusion is

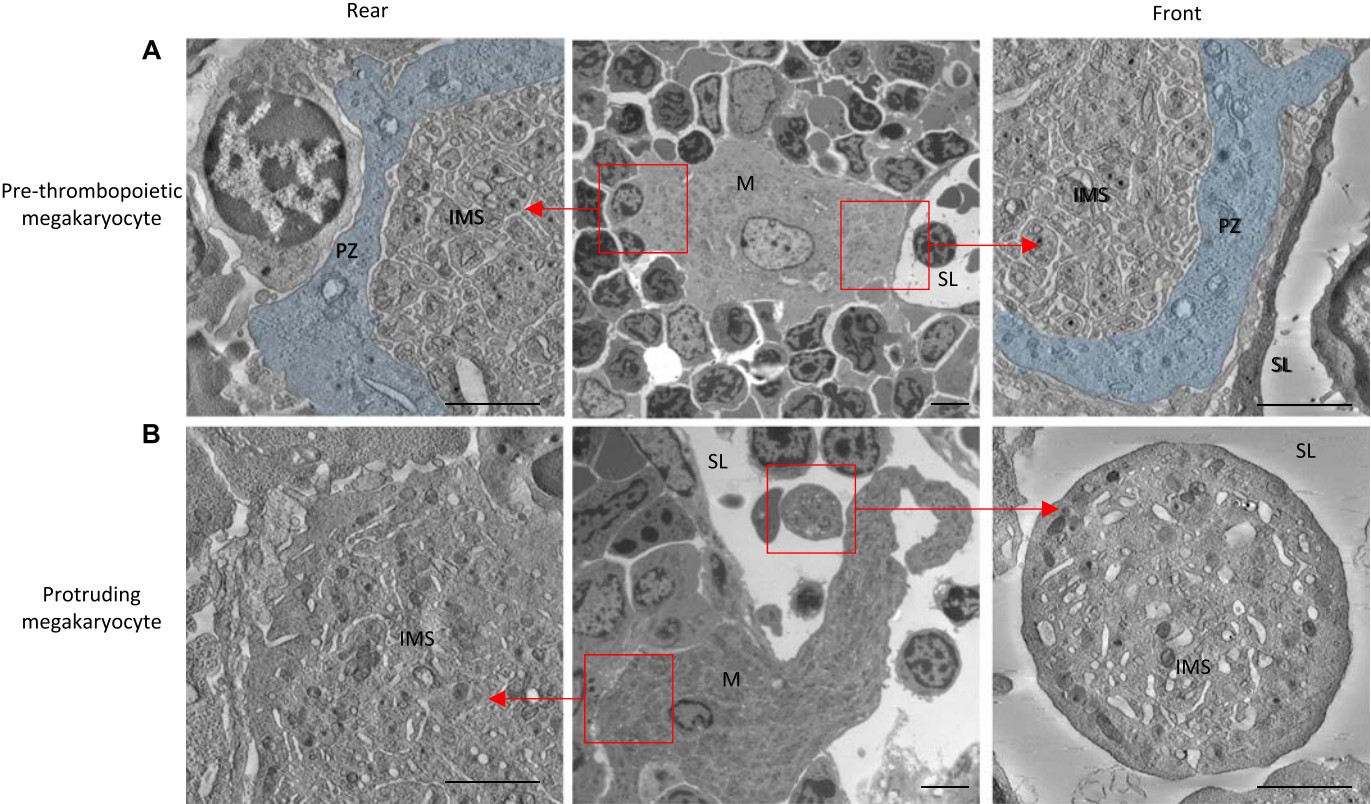

Rear Front

**Figure 5. The peripheral zone (PZ) is lost in protruding megakaryocytes.**
**(A)** Two regions (red boxes) selected for tomography from a non-protruding megakaryocyte (middle panel) with normal PZ (blue) and IMS at both the rear (left panel) and front (right panel) of the cell. **(B)** Two regions (red boxes) selected for tomography from a protruding megakaryocyte (middle panel) lacking a PZ and allowing the IMS closer to the plasma membrane at both the rear (left panel) and front (right panel) of the cell. Bone marrow of the diaphysis region obtained from flushed femurs. Scale bars represent 2 μm. Images representative of three independent experiments.
Source data are available for this figure.

likely to be the mechanism by which protrusion occurs. The combination of intravital light microscopy and correlative EM thereby reveals a novel mechanism for the intravasation of megakaryocyte and the generation of blood platelets in vivo.

## Megakaryocytes predominantly extend into sinusoids as large protrusions, not as proplatelets

Using intravital CLEM and large-volume electron tomography, we put forward a novel mechanism for the intravasation of megakaryocyte material that ultimately leads to the generation of blood platelets. The current model, which has dominated the field for more than 40 years, suggests that megakaryocytes extend proplatelets from the cell body into bone marrow sinusoids using force exertion from MT sliding (12). The recent study from Lefrançais et al (16), however, shows that platelets are generated from megakaryocyte material that enters the vasculature as much larger fragments or complete cells. Our data, however, seem potentially to contradict those of previous studies, where proplatelet-like structures have been described to be entering sinusoids and visualized by intravital fluorescence imaging, e.g., as shown in Junt et al (5). This is logical because their length, diameter, morphology, and the dynamics of extension/retraction appear similar to those of proplatelets seen in vitro. However, using an intravital CLEM

technique in this study has allowed us for the first time to observe these structures directly in vivo in combination with high-resolution TEM analysis. The considerably enhanced level of magnification and detail afforded by this approach revealed marked differences in both morphology and cytoskeletal organization when comparing in vitro– and in vivo–formed megakaryocyte extensions. So, although we observed proplatelets to be abundant in marrow sinusoids, they are mostly found free within the sinusoids and the primary form of megakaryocyte extension is a large protrusion. We, therefore, suggest that classical proplatelets predominantly represent a later stage in thrombopoiesis, one that occurs after release of a large protrusion into the sinusoid lumen. Culturing megakaryocytes in vitro and seeding them on to fibrinogen-coated surfaces seem to accelerate the process, bypassing the large protrusion stage. This may be a crucial factor contributing to the difficulty in generating functional platelets in vitro.

## Large protrusions do not extend via MT sliding: investigating the mechanism of megakaryocyte intravasation

In addition to the structural differences between megakaryocyte large protrusions and proplatelets, the mechanism of extension also differs. The thick marginal band of overlapping MTs within the tip of proplatelets lies just under the membrane, consistent with them

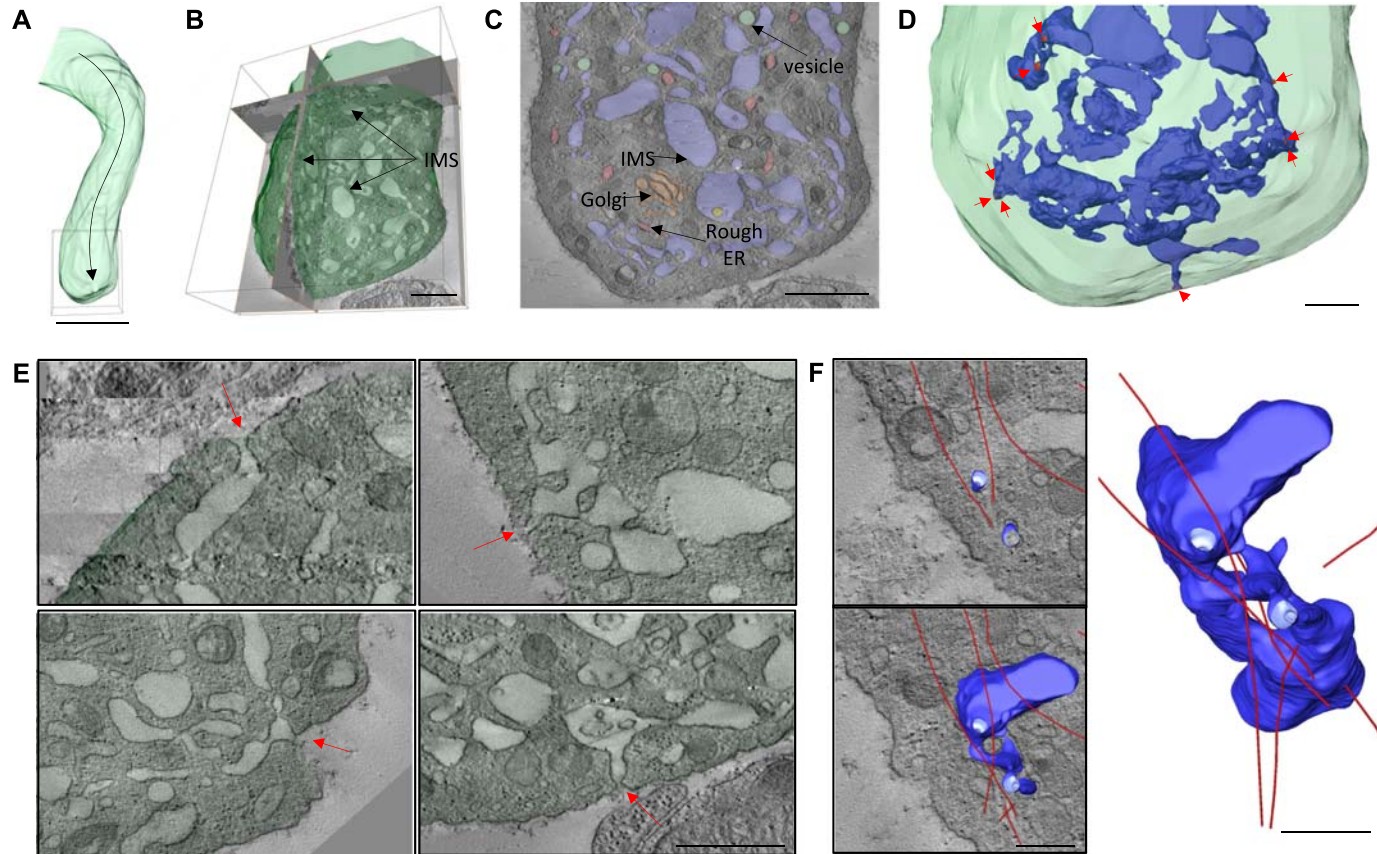

**Figure 6. The IMS fuses to the plasma membrane at the protrusion tip.**
**(A)** Area of protrusion (box) selected for large-volume tomography. **(B)** Large-volume tomographic reconstruction of the protrusion tip showing the IMS throughout. **(C)** Tomographic section showing different membrane types. **(D)** Model showing multiple sites of fusion (arrows) between the IMS (blue) and the plasma membrane. **(E)** Tomographic slices of selected fusion sites. **(F)** Model showing the relationship between a partial reconstruction of the IMS (blue) and MTs (red) at a fusion site. Bone marrow of the diaphysis region obtained from flushed femurs. Scale bars represent 5 µm in (A); 1 µm in (B, C); and 500 nm in (D–F). Images representative of three independent experiments.

transmitting force to the membrane to promote extension. In contrast, however, large-volume tomography undertaken in this study showed that there was substantially less MT bundling at the tip of large protrusions, and the MTs were not found in an immediately sub-plasma membrane location. This makes it unlikely that they are responsible for extension of the large protrusion. In addition, tomography revealed no extensive actin cytoskeleton at the leading edge and rear of protruding megakaryocytes, suggesting that actin polymerization is also not responsible for migration through the endothelium. Instead, we propose a distinct mechanism for intravasation of megakaryocyte material in vivo, involving fusion of internal membranes with the plasma membrane.

### Megakaryocytes extend into sinusoids by mass fusion of their internal membrane with the plasma membrane

Non-protruding megakaryocytes possess a tightly packed, intact "peripheral zone," limiting contact between the IMS and the plasma membrane (Fig 8A). Breakdown of the peripheral zone by an as yet unknown mechanism allows close contact and fusion between these membrane systems. After penetration of the endothelium and anchorage to the luminal surface (Fig 8B), the megakaryocyte extrudes internal membrane into the plasma membrane at the leading edge using the numerous points of fusion (Fig 8C). The anchorage to the endothelium allows forward progression at the leading edge of the protrusion and prevents it from slipping back inside the bone marrow parenchyma. This tethering of the membrane would also suggest that the protrusion is not free to migrate through the opening in the endothelium but must instead extend into the sinusoid lumen by extruding membrane at the protrusion tip. From our tomographic reconstructions, we estimate that the IMS provides a potential fivefold increase in membrane surface area. Extrusion of internal membrane at multiple points of fusion cause a local increase in membrane surface area and allow the tip to extend into the sinusoid lumen (Fig 8C). This represents a novel mechanism of cell motility, one which may effectively be unique to megakaryocytes because of their large amounts of internal membrane. Once sufficiently extended into the sinusoid lumen, the protrusion is detached by a mechanism that is currently unclear, and although shear forces associated with blood flow may be important, it is also likely that there is also a regulated cellular mechanism. The progression from a free large protrusion to proplatelet/proplatelets and ultimately platelets (Fig 8D) requires further study. Clearly, it requires a dramatic reorganization of both internal membrane and MTs, followed by multiple scission steps, which, given the regular size of platelets, is also likely to be tightly regulated.

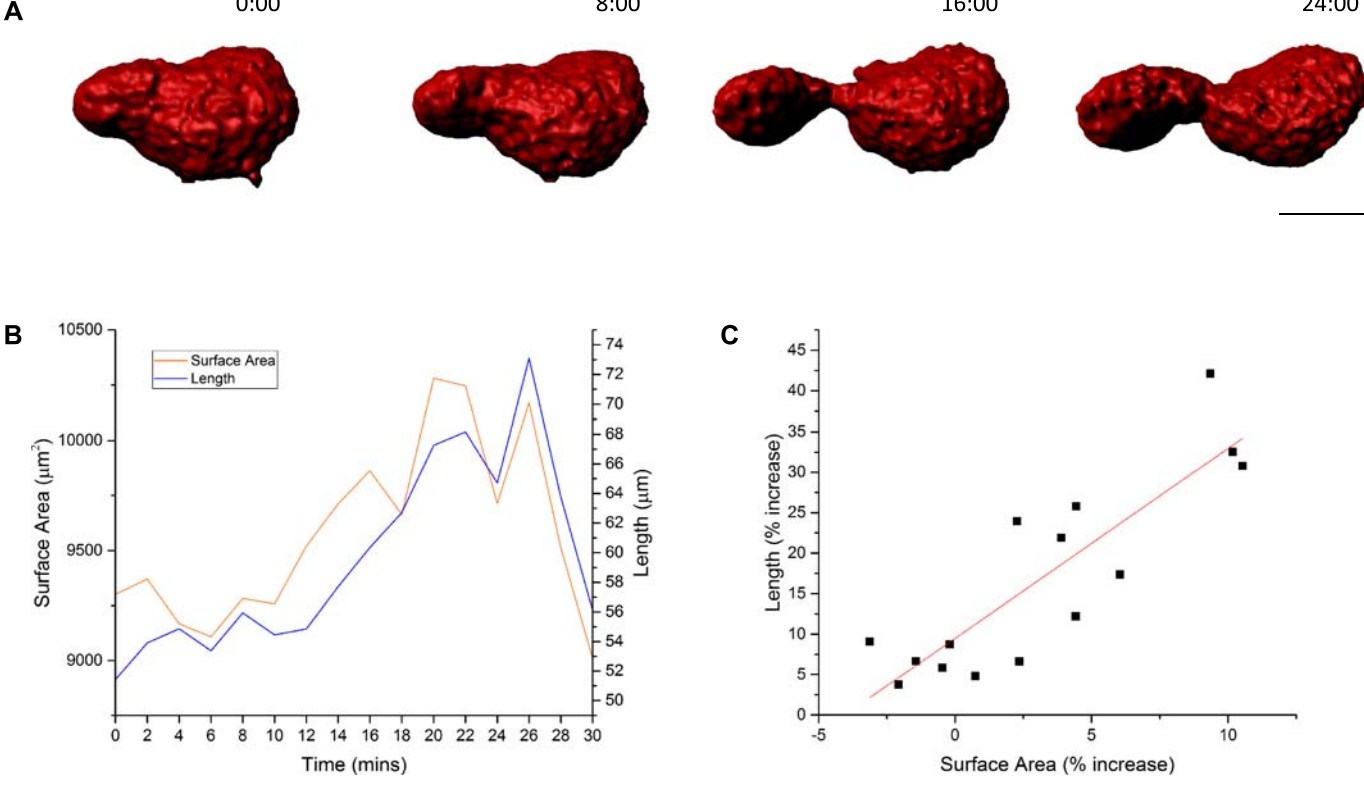

**Figure 7. Megakaryocyte total surface area increases with protrusion extension.**
**(A)** Selected images of model generated from time-lapse intravital z-stacks of a megakaryocyte protrusion extending to the left into a sinusoid (not shown). **(B)** A graph of megakaryocyte surface area and protrusion length over time. **(C)** A dot plot of protrusion length against surface area. Images obtained from calvarium bone marrow. Scale bar represents 20 μm. Data representative of four independent experiments.

In summary, intravital CLEM, as described here, challenges our current view of how platelets are generated, through migration of megakaryocyte protrusions into the vasculature. The approach has allowed us to observe dynamic events in unprecedented detail within the bone marrow in real-time using the ultrahigh-resolution of TEM and revealed that mass membrane fusion drives the extension of large protrusions into marrow sinusoids, in contrast to a MT-driven proplatelet formation. Although proplatelets are also abundant in marrow sinusoids, the detailed approach afforded by intravital CLEM makes it clear that most of these are not directly connected with megakaryocytes within the marrow parenchyma, and their formation is, therefore, likely to come from large protrusions after exit of these structures into the blood vessel. The power of a combined light and electron microscopic analysis will undoubtedly also prove to be an invaluable tool in the study of ultrastructural details of a variety of other elements and mechanisms.

## Materials and Methods

### Mice

C57/Bl6, age- and sex-matched bone marrow recipient mice were purchased from Harlan UK. mTmG mice were purchased from Jackson laboratories. For vWF-tdTomato mice, a tdTomato cDNA was inserted into the RP23-382D6 BAC (https://bacpacresources.org) at the initiation codon of the Vwf coding sequence within exon 2 by homologous recombination. A polyadenylation signal was included at the end of tdTomato cDNA. Circular BAC DNA was prepared using the QIAGEN Large-Construct Kit (QIAGEN) and injected into hybrid C57Bl6-CBA oocytes by intra-cytoplasmic sperm injection (22) using C57Bl6 sperm.

Whole bone marrow mononuclear cells from mTmG and vWF-tdTomato mice were harvested from femurs and $10^6$ cells were injected intravenously into each lethally irradiated recipient (10 Gy, over two doses 3+ hours apart). The resulting chimeric mice were analyzed by intravital microscopy 8–14 weeks later.

Fetal liver megakaryocytes for in vitro study were prepared from C57/Bl6 mice as previously described (23). All animal studies were approved by local research ethics committees (AWERB) and licensed under UK Home Office project licenses PPL 30/2908 and 70/8403.

### Platelet depletion

Platelet numbers were depleted by intraperitoneal administration of anti-GPIbα antibody (emfret Analytics) at a dose of 2 μg/g bodyweight. Intravital imaging was performed 4 days after when the rate of platelet production should have been maximal.

### In vitro CLEM

Megakaryocytes were isolated from mouse fetal livers and enriched with a 1.5%–3% albumin gradient as previously described (23).

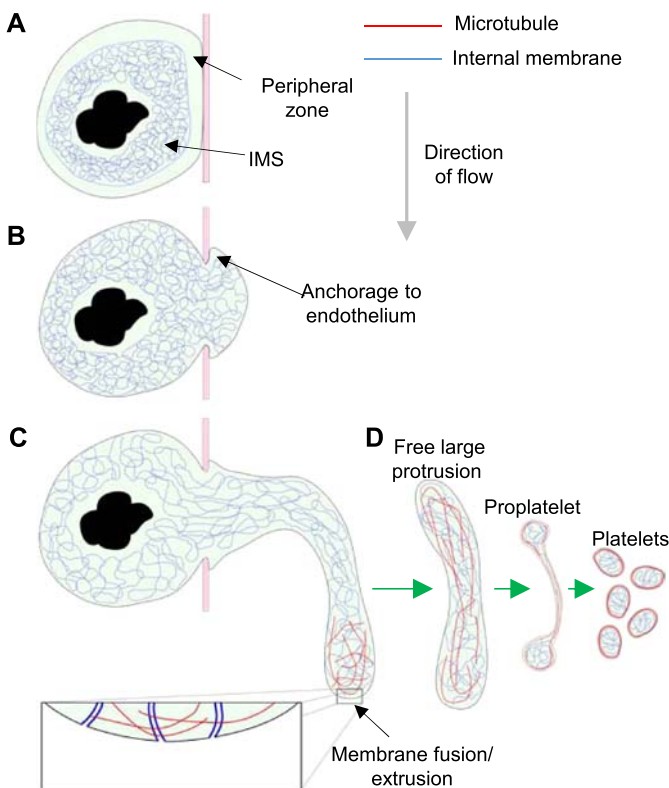

**Figure 8.  Diagrammatic representation of thrombopoiesis in vivo.**
**(A)** Non-protruding megakaryocyte with intact peripheral zone and densely packed IMS. **(B)** Loss of peripheral zone allows IMS to fuse with plasma membrane and a protrusion to form, which anchors to the luminal surface of the endothelium. **(C)** IMS trafficked along MTs continues to extrude into the protrusion plasma membrane resulting in extension into the sinusoid lumen in the direction of flow. After release of the large protrusion, MT and membrane reorganization must occur to form proplatelets and ultimately platelets.

Megakaryocytes were added to fibrinogen-coated (100 μg/ml) live cell imaging dishes (Matek), which possess a raised grid pattern in the glass. Proplatelet formation was observed using DIC imaging at 37°C with 5% $CO_2$ overnight. Sample processing for TEM was performed as previously described (24). Briefly, cells were fixed initially with 2.5% glutaraldehyde in 0.1 M sodium cocodylate buffer (pH 7.4), followed by 1% osmium tetroxide. Cells were stained with 1% uranyl acetate and dehydrated through a graded series of ethanol before embedding in Epon resin and hardening at 60°C overnight. The coverslip was removed from the hardened Epon by submerging in liquid nitrogen followed by plunging into boiling water. The resin was trimmed down to the area imaged by DIC and sectioned to 250 nm.

### Intravital microscopy

Intravital microscopy was largely performed as described previously (25, 26). Briefly, mice expressing either mTmG-Tomato or vWF-tdTomato were anaesthetized using 3% isoflurane in 2 l/min medical oxygen and then maintained at 1%–2% isoflurane during the experiment. Once surgical anesthesia was obtained, the mice were transferred to a heat mat to maintain body temperature and the skin covering the top of the skull was removed carefully. Dental

cement was used to affix a custom-made frame to the skull to allow the mice to be positioned in a custom-made stage insert and stabilize the skull for imaging. Calvarium bone marrow was imaged by performing z-stacks by single-photon laser-scanning confocal microscopy on an upright Leica TCS SP5 or Zeiss 780 NLO microscope using a long working distance 25× 0.95 NA or 20× 0.1 NA, respectively, water immersion objective with the confocal pinhole set to give an optical section 3–4-μm thick. Surface models of protruding megakaryocytes were generated from z-stack time-lapse imaging using Imaris.

### Intravital CLEM

A flow diagram–based description of the process is illustrated in Fig S2 and is as follows. To observe megakaryocytes within the bone marrow, lethally irradiated wild-type recipient mice were reconstituted with either mTmG (as shown in Fig S2) or vWF-tdTomato whole bone marrow in which either all cells or just those of the megakaryocyte lineage express the red fluorescent protein tomato and analyzed using confocal intravital microscopy of calvarium bone marrow (26, 27) (Fig S2A). To delineate the vasculature, the mice were injected intravenously with 50 μg anti-glycoprotein 1b AlexaFluor488 (X488; emfret Analytics) and 8 μg Cy5-dextran (CX500-S5-1, 500 kD; Nanocs) in PBS approximately 10 min before commencing imaging. While imaging the bone marrow within the calvarium, a megakaryocyte was observed entering a sinusoid, as shown by glycoprotein 1b labeling within the vasculature being continuous with mTmG-tomato signal within the bone marrow parenchyma. This event was captured by taking a series of z-stacks over a 30-min period (Fig S2B). Acquisition of a larger area of bone marrow (Fig S2C) showed the location of the protrusion observed within the context of the surrounding tissue. This large 3D image acted as a reference for locating the area of interest to be trimmed and sectioned for TEM analysis. After intravital microscopy, the mice were euthanized by cervical dislocation and the top of the skull was removed and rapidly fixed by submerging in 2.5% glutaraldehyde and 2% paraformaldehyde in 0.1 M sodium cocodylate buffer (pH 7.4) at 37°C (Fig S2D). This method was chosen to achieve the best possible correlation between the intravital and EM data, as perfusion fixation could result in changes to the protrusion from the final frame of intravital imaging because of the additional flow of fluid through the vasculature required for this approach. The thickness of the bone in the region of imaging was 30 μm, allowing the fixative to penetrate the marrow tissue rapidly. After fixation, the skulls were decalcified in 10% EDTA in PBS (pH 7.4) at 4°C for 7 days, changing the solution every 48 h. The skulls were further fixed in 1% osmium tetroxide and stained with 1% uranyl acetate before dehydration through a graded series of ethanol. The skulls were washed three times with propylene oxide and left rotating overnight in a 1:1 mixture of propylene oxide and Epon. Propylene oxide was allowed to evaporate over 4 h before the skulls were transferred into fresh Epon and left rotating overnight. The skulls were positioned in molds and left to harden at 60°C for 48 h (Fig S2E). Once processed for TEM, the vasculature was easily visible under a dissecting microscope (Fig S2F) and the distinctive branching structure could be readily correlated with the reference intravital image. The area containing the protrusion was trimmed to a convenient size for serial sectioning (Fig S2G) on an ultra-microtome. To locate the area of interest in the z-axis, thick sections (1 μm) were taken

and viewed on a light microscope (Fig S2H) until ~5 μm above the desired region, at which point serial semi-thick sections (300 nm) were taken (Fig S2I). A low-magnification TEM image (Fig S2J) confirmed that the area observed by fluorescence microscopy had been found. The sections containing the protrusion were identified by comparing the vascular architecture in TEM images with that of z-stacks taken during fluorescence imaging. Once the sections of interest had been determined, serial high-magnification TEM images of the protrusion were taken and aligned to reconstruct the 3D volume (Fig S2K and L, and Video 3). From these images, structures of interest were segmented (Fig S2M) and a model was constructed (Fig S2N–P). The model revealed that most of the megakaryocyte was within the sinusoid lumen, with only a small region still within the bone marrow parenchyma. The cell body of the megakaryocyte consisted mainly of nucleus, most of which had also crossed into the sinusoid lumen. This megakaryocyte was clearly in the final stages of intravasation; almost all its cytoplasm had been released into the circulation and the remainder of the cell (mostly nuclear material) was being ejected from the bone marrow parenchyma.

Although focused ion beam/scanning EM or serial block face/scanning EM would also have allowed us to reconstruct bone marrow volumes, these two approaches destroy the material soon after acquiring the image (28, 29). When the reconstructed volumes are subsequently analysed, any regions of interest which may have warranted further investigation have been lost. Using conventional serial sectioning as we have done, preserves the material so that tomography can be carried performed (Fig S2Q) to reveal ultrastructural details (Fig S2R and S), which are beyond the resolution limit of standard SEM.

### Preparation of femoral bone marrow

Mice were perfusion-fixed with 2.5% glutaraldehyde and 2% para-formaldehyde in 0.1 M sodium cocodylate buffer warmed to 37°C. Femurs were removed and immersion-fixed for a further 1 h before both ends of the femur were cut and the marrow was flushed out as a whole piece of tissue by gently flowing through the fixative using a 21-gage needle as previously described (30). This was immersed in a fresh fixative for an additional hour. The marrow was postfixed, stained, and embedded in Epon as described above.

### Histology

Bone marrow was fixed, removed, and processed as described above. Serial sections of whole bone marrow were cut on a glass knife at a thickness of 2 μm and collected on slides. The sections were stained with toluidine blue. Stitched tile scans of entire sections were acquired on a Leica DMI6000 inverted microscope using a 40× 0.6 NA air objective.

### EM/tomography

TEM images which were used to construct whole cell models of megakaryocytes were acquired on a Tecnai 12 (FEI, 120 kV) electron microscope. For tomography, the sections were standardly cut to a thickness of 300 nm. Tilt series were performed on selected areas of interest using a Tecnai 20 (FEI, 200 kV) electron microscope.

Images were acquired every 1° of tilt over a range of ±40° and every 0.5° beyond to a maximum tilt angle of ±70°.

### Image alignment

Alignment of histological and TEM images was performed using Fiji's registration plugin. The TEM image stacks were segmented and 3D models were generated using Amira software.

### Tomographic reconstructions

Tomograms were reconstructed in 3D using eTomo within IMOD (Boulder Laboratory). ETOMO was also used to align and join multiple tomographic reconstructions. 3D models were created using Amira 6.0.0 software. Center lines of MTs were traced using the XTracing extension within Amira 6.0.0. Continuity of MTs between adjoining reconstructions was determined manually.

### Statistics

Statistical analysis was performed using Origin 2016. Statistical significance of megakaryocyte protrusions was assessed using an unpaired $t$ test. $P < 0.05$ was considered statistically significant.

### Data availability

Source data for tomogram reconstruction are available in the BioStudies database (http://www.ebi.ac.uk/biostudies) under the accession number S-BSST147.

# Supplementary Information

# Acknowledgements

This work was supported by the British Heart Foundation (Programme grants RG/10/006/28299 and RG/15/16/31758) and the European Research Council (STG 337066). LM Carlin is supported by the Medical Research Council (grant MR/M01245X/1), the National Heart and Lung Institute Foundation, and Cancer Research UK. Intravital microscopy was in part performed at the Imperial College Facility for Imaging by Light Microscopy and in part supported by funding from the Wellcome Trust (grant P49828) and the Biotechnology and Biological Sciences Reseach Council (grant P48528). We are grateful to Dr. Paul Verkade for valuable discussion of the work. We also thank G Tilly, J Mantell, A Leard, and S Cross (Wolfson Bioimaging Facility, University of Bristol).

## Author Contributions

E Brown: conceptualization, data curation, formal analysis, investigation, visualization, methodology, and writing—original draft.
LM Carlin: resources, data curation, investigation, methodology, and writing—review and editing.
C Nerlov: resources and writing—review and editing.

C Lo Celso: resources, data curation, methodology, and writing—review and editing.

AW Poole: conceptualization, resources, supervision, funding acquisition, visualization, project administration, and writing—review and editing.

**Conflict of Interest Statement**

The authors declare that they have no conflict of interest.

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
