## [Reviewer comments · Life Science Alliance]

Multiple membrane extrusion sites drive megakaryocyte migration into bone marrow blood vessels

Edward Brown, Leo M. Carlin, Claus Nerlov, Cristina Lo Celso and Alastair W. Poole
DOI: 10.26508/lsa.201800061

Review timeline:	Submission Date:	29 March 2018
	Revision Received:	29 March 2018
	Editorial Decision:	9 April 2018
	Accepted:	27 April 2018

Report:

(Note: Letters and reports are not edited. The original formatting of letters and referee reports may not be reflected in this compilation.)

Please note that the manuscript was previously reviewed at another journal and the reports were taken into account in the decision-making process at Life Science Alliance. Since the original reviews are not subject to Life Science Alliance's transparent review process policy, the reports and author response cannot be published.

1st Editorial Decision

9 April 2018

Thank you for submitting your manuscript entitled "Megakaryocytes migrate into murine bone marrow blood vessels by extrusion of internal membrane" to Life Science Alliance. Your manuscript has been reviewed at another journal, and you provided a revised version and a point-by-point response to the previous round of review to us.

Two Life Science Alliance editors have read your manuscript and assessed it in light of these previous reports. You use intravital correlative light electron microscopy to show platelet formation/megakaryocyte migration into the sinusoidal space via large protrusion formation. We value your work and the revision performed, and we appreciate the way you addressed the previous referees' concerns regarding the data presentation and lack of quantifications. We still have some suggestions on how to improve the discoverability and re-use of your data, but we are happy to accept your manuscript for publication in Life Science Alliance should you be able to address these suggestions:

Please provide source data for all EM images to allow re-analysis of your work by others. Please also use bigger EM images in figures 3 and 6 (6d and 6e), so that the ultrastructures you describe can be more easily appreciated. We would like to furthermore propose to add an overview image for a better spatial understanding of what is shown in figure 4b and 4c as well as in figure 5. The latter would ideally display the whole cell with insets for front and rear parts that are individually displayed subsequently. The supplementary movie #2 should also be mentioned in conjunction with figure 6 in our view.
